# Stable and efficient pure blue quantum-dot LEDs enabled by inserting an anti-oxidation layer

Wenjing Zhang[1,4], Bo Li[2,4], Chun Chang[3,4], Fei Chen[1] ✉, Qin Zhang[3], Qingli Lin[1], Lei Wang[1], Jinhang Yan[1], Fangfang Wang[1], Yihua Chong[1], Zuliang Du ⬤[1], Fengjia Fan ⬤[2] ✉ & Huaibin Shen ⬤[1] ✉

The efficiency and stability of red and green quantum-dot light-emitting diodes have already met the requirements for commercialization in displays. However, the poor stability of the blue ones, particularly pure blue color, is hindering the commercialization of full-color quantum-dot light-emitting diode technology. Severe hole accumulation at the blue quantum-dot/hole-transport layer interface makes the hole-transport layer prone to oxidation, limiting the device operational lifetime. Here, we propose inserting an anti-oxidation layer (poly(p-phenylene benzobisoxazole)) between this interface to take in some holes from the hole-transport layer, which mitigates the oxidation-induced device degradation, enabling a $T_{50}$ (time for the luminance decreasing by 50%) of more than 41,000 h with an initial brightness of 100 cd m$^{-2}$ in pure blue devices. Meanwhile, the inserted transition layer facilitates hole injection and helps reduce electron leakage, leading to a peak external quantum efficiency of 23%.

With excellent color rendering and cost-effective solution processability, quantum-dot light-emitting diodes (QD-LEDs) are attractive light sources for next-generation solid-state lighting and display technologies[1–10]. Recently, the external quantum efficiency (EQE) of green- and red-QD-LEDs have surpassed 25%, and the $T_{50}$ and $T_{95}$ (@100 cd m$^{-2}$) lifetimes (defined as the time for luminance dropping to 50% and 95% of the initial value ($L_0$)) have exceeded 2,000,000 h and 500,000 h, respectively, which are comparable to the state-of-the-art organic LEDs (OLEDs) and have met the requirements for display applications[9,11–14]. However, for blue QD-LEDs, particularly pure blue ones, although the state-of-the-art $T_{50}$ lifetime (@100 cd m$^{-2}$) of sky blue (466-480 nm) QD-LEDs has surpassed 50,000 h[3], the $T_{50}$ lifetime for pure blue (455-465 nm) QD-LEDs is only 15,850 h[15], falling short of the criteria for display applications. Therefore, developing stable and

high-efficiency pure blue QD-LEDs is crucial for promoting the development of QD-based display technology.

In blue QD-LEDs, there are considerable hole injection barriers[6–24], which lead to severe hole accumulation and make hole-transport layers (HTLs) prone to oxidation. Therefore, exploiting an effective strategy to mitigate oxidation-induced damage is the key to addressing the poor stability issue of pure blue QD-LEDs. Here, we introduce an anti-oxidation transition layer poly(p-phenylene benzobisoxazole) (PBO) (highest occupied molecular orbital (HOMO), −5.88 eV; lowest unoccupied molecular orbital (LUMO), −2.70 eV; hole mobility $\mu_h$ ~ $1.14 \times 10^{-3}$ cm$^2$ V$^{-1}$ s$^{-1}$) between the QD layer and the HTL. This transition layer takes in some holes from the HTL and itself is less prone to oxidation, allowing us to fabricate stable pure blue QD-LEDs. Meanwhile, thanks to its deeper HOMO level, the hole injection is

[1]Key Laboratory for Special Functional Materials of Ministry of Education, National & Local Joint Engineering Research Center for High-efficiency Display and Lighting Technology, Henan University, 475004 Kaifeng, China. [2]Hefei National Laboratory for Physical Sciences at the Microscale and Department of Modern Physics, CAS Key Laboratory of Microscale Magnetic Resonance, Synergetic Innovation Center of Quantum Information and Quantum Physics, University of Science and Technology of China, 230026 Hefei, China. [3]Key Laboratory of Nondestructive Testing Ministry of Education, Nanchang Hangkong University, 330063 Nanchang, China. [4]These authors contributed equally: Wenjing Zhang, Bo Li, Chun Chang. ✉e-mail: chenfei.henu@henu.edu.cn; ffj@ustc.edu.cn; shenhuaibin@henu.edu.cn

improved, leading to more efficient radiative recombination and improved peak EQE. This strategy allows us to fabricate QD-LEDs with peak EQEs of 23% and long operational lifetimes of more than 41,000 h at 100 cd m$^{-2}$ - both are state-of-the-art values yet reported for solution-processed pure blue QD-LEDs.

## Results

### Device design by inserting an anti-oxidation transition layer

To improve the stability and efficiency of blue QD-LEDs, continuous efforts have been devoted to modifying the emitting layer (EML), constructing the QD structure, as well as optimizing the transport layer[15,23–30]. However, pure blue devices with simultaneous high stability and high EQE are yet to be demonstrated[15,28]. As shown in Fig. 1a and Supplementary Fig. 1, due to the deep valence band energy level of blue QDs, there is a large injection barrier between the QD and the HTL, causing remarkable hole accumulation in HTL. To investigate the potential effect of hole accumulation on HTL, we performed cyclic voltammetry characterizations on TFB, the most commonly used HTL, and found they are prone to be oxidized. Therefore, we sought to

reduce hole accumulation in HTL or transfer holes to an inserting layer that is less prone to oxidation. We hypothesized PBO can be a good candidate (Fig. 1b), because when we increase the cyclic voltammetry scanning time, the change in oxidation-reduction potentials of PBO film is smaller than that of TFB film, indicating it has higher electrochemical stability under positive bias. Moreover, PBO has excellent heat resistance and stability (Supplementary Fig. 2). In previous reports, PBO has been used as an electron-blocking layer between the QD layer and the electron transport layer (ETL)[31–34] to block excess electron injection. But less is realized that PBO has a deep valence band (−5.88 eV, 0.52 eV deeper than TFB) and a high hole mobility ($\mu_h \sim 1.14 \times 10^{-3}$ cm$^2$ V$^{-1}$ s$^{-1}$)[35–37] (Fig. 1c and Supplementary Fig. 3). Therefore, they are also suitable for hole injection/transport. If they are used as an inserting layer between TFB and QD, there will be less hole injection barrier and hole concentration in TFB can be reduced.

To verify whether this is the case experimentally, we fabricated the pure blue devices using a solution-processed organic-inorganic hybrid QD-LED architecture: ITO/PEDOT:PSS (18 nm)/TFB (18 nm)/PBO (10 nm)/QDs (32 nm)/ZnMgO (50 nm)/Al (100 nm) (Supplementary

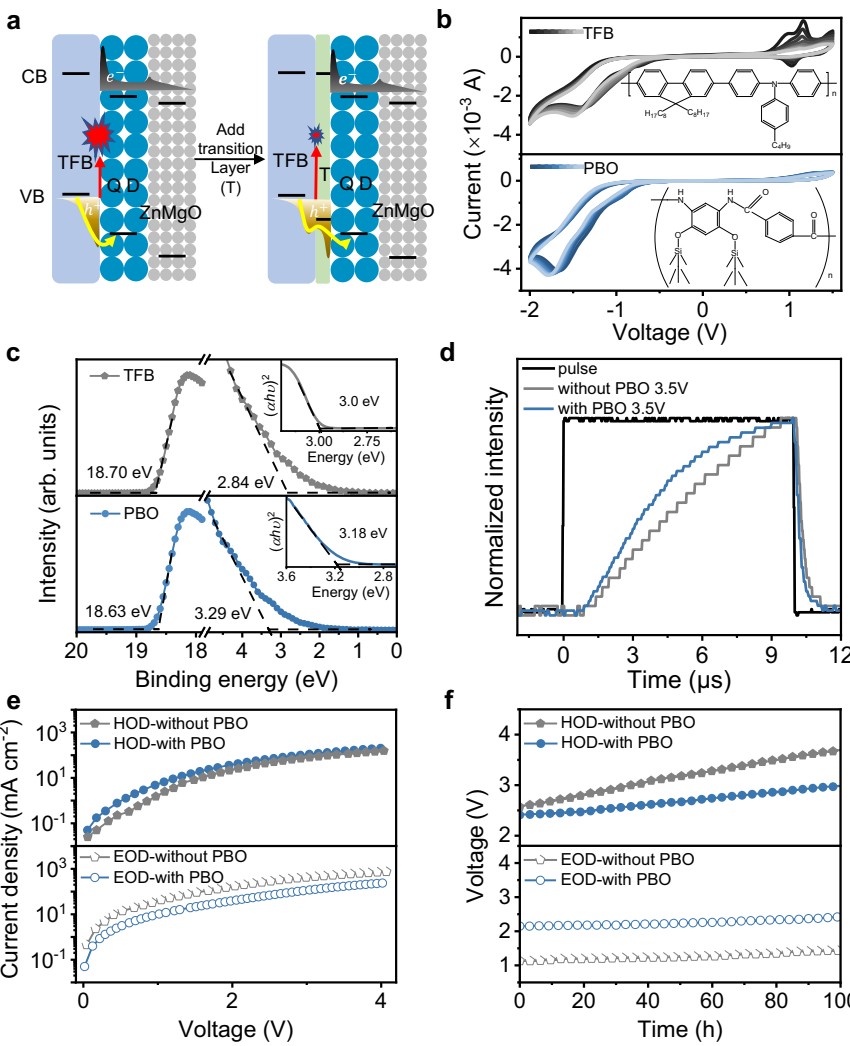

**Fig. 1 | The benefits of introducing a PBO transition layer. a** Schematic illustration of damage to the hole-transport layer (HTL) caused by the carrier accumulation without and with a transition layer (T). **b** The cyclic voltammetric curves of poly[(9,9-dioctylfluorene-2,7-diyl)-co-(4,4′-(N-(p-butylphenyl))diphenylamine)] (TFB) and poly(p-phenylene benzobisoxazole) (PBO). The insets are the chemical structures of TFB and PBO. **c** The ultraviolet photoelectron spectroscopy (UPS) spectra of the high-binding energy secondary electron cutoff regions and the valence-band edge regions of TFB and PBO. The inset is the Tauc plot of TFB and

PBO between ($\alpha$h$\nu$)$^2$ and photon energy. **d** Transient electroluminescence (tr-EL) spectra for the quantum-dot light-emitting diodes (QD-LEDs) under 3.5 V without and with PBO. **e** The current density-voltage characteristic curves of hole-only-device (HOD) with ITO/PEDOT:PSS/TFB/QDs/MoO$_3$/Al, ITO/PEDOT:PSS/TFB/PBO/QDs/MoO$_3$/Al, and electron-only-device (EOD) with ITO/ZnMgO/TFB/QDs/ZnMgO/Al, ITO/ZnMgO/TFB/PBO/QDs/ZnMgO/Al. **f** Operation voltage versus time of HOD and EOD devices without and with PBO operated at a constant current.

Fig. 4a, b, see "Methods" section for the fabrication and characteristics of the QD-LEDs). We used CdZnS/ZnS core/shell QDs with narrow luminescent peak (FWHM ~ 26 nm) around 455 nm and high absolute photoluminescence quantum yield (PL QY) of almost 100% as an emitting layer[25] (see Supplementary Fig. 5 for detail). We measured transient electroluminescence (tr-EL) dynamics, which reflects the hole injection dynamics because the injection barrier and effective mass of the electron are far less than that of the hole in II-VI QDs. We observed that in devices with an inserting PBO layer, the EL indeed rises faster than those without PBO (Fig. 1d and Supplementary Fig. 6a, d), suggesting that the PBO layer promotes hole injection.

We also fabricated hole-only-device (HOD: ITO/PEDOT:PSS/TFB/QDs/MoO$_3$/Al, ITO/PEDOT:PSS/TFB/PBO/QDs/MoO$_3$/Al) and the electron-only-device (EOD: ITO/ZnMgO/TFB/QDs/ZnMgO/Al, ITO/ZnMgO/TFB/PBO/QDs/ZnMgO/Al) (See Supplementary Fig. 7 for energy level diagrams), the current density-voltage curves also show that the addition of PBO promotes hole injection at low driving voltages and reduces the electron leakage (Fig. 1e). To investigate whether PBO offers the desired protection to TFB, we measured voltage variations of HOD and EOD at a constant current density of 50 mA/cm². The driving voltage of both EOD devices without and with PBO (Fig. 1f) slightly increases by about 0.30 V in 100 h. Contrastingly, voltage increases by 1.13 V in 100 h for the HOD without PBO, and with PBO, voltage only increases by about 0.57 V in 100 h. At the same time, the QDs films with TFB and PBO exhibit similar surface roughness according to the atomic force microscopy (AFM) images (Supplementary Fig. 8), suggesting the difference in voltage change is not related to film quality. We conclude that device degradation is more likely related to damage in hole injection[12,38,39], and PBO can serve as an anti-oxidant protection layer for hole injection as expected. In addition, the quenching of QD emissions by the TFB HTL can also be significantly suppressed by introducing the PBO layer (Supplementary Fig. 9).

## Device characterizations

We then performed further performance characterization on QD-LED. They exhibit a saturated pure blue color (Fig. 2a) with corresponding Commission Internationale de l'Eclairage (CIE) chromaticity

coordinates of (0.146, 0.040) (Supplementary Fig. 10). The peak EQE surpasses 20% when we insert a 10 nm PBO layer (Supplementary Fig. 11). According to the current density-luminance-voltage characteristics for the QD-LEDs with and without PBO (Fig. 2b), the QD-LEDs with PBO present a slightly lower injection current before turning on, indicating a smaller leakage current. Although those two devices show a similar turn-on voltage (3.2 V), the brightness in QD-LEDs with PBO is about 2.6-fold that of the device without PBO under the same current density (~5.8 V, ~287 mA cm⁻²). With improved luminance but lower current density, the EQE of QD-LEDs with PBO reaches a value of 23%, which is among the best-performing reported values and is about two times higher than that of the device without PBO (Fig. 2c), making this device state-of-the-art solution-processed pure blue QD-LEDs so far (Supplementary Table 1). The histogram in Fig. 2d shows an average peak EQE of 20.8% with a low standard deviation of 2.3%, suggesting high reproducibility. The above results indicate that the introduction of PBO can effectively suppress leakage current and improve electroluminescence (EL) efficiency.

## Device stability tests

The operational lifetime of QD-LEDs is closely related to the stability of organic HTLs, we expect less damage in HTL enabled by PBO will improve the operational lifetime of the devices. In our stability test, devices with and without PBO were driven at an initial $L_0 = 3914$ cd m⁻² and $L_0 = 1770$ cd m⁻² with a constant current density of 137 mA cm⁻² and 166 mA cm⁻² (Fig. 3a), respectively. The device without PBO drops to 50% of its initial value after about 30 h, equivalent to 4995 h at 100 cd m⁻² with an empirical acceleration factor of n = 1.78, which is obtained by fitting $T_{50}$ values at various luminance values (Fig. 3b). Devices with PBO buffer layers are much more stable, and we obtained a maximum $T_{50}$ lifetime of 41,022 h at the initial luminance of 100 cd m⁻², corresponding to a $T_{95}$@100 cd m⁻² of ~5264 h or a $T_{95}$@1000 cd m⁻² of ~87 h. This is the most stable pure blue QD-LED reported to date and the stability approaches that of state-of-the-art sky blue QD-LEDs (Supplementary Table 1). The histograms of the lifetime of 25 devices with and without PBO (Fig. 3c) reveal excellent fabrication reproducibility, and the devices with PBO exhibit an average $T_{50}$ lifetime at 100 cd m⁻² of over 31,000 h, contrasting to 4600 h

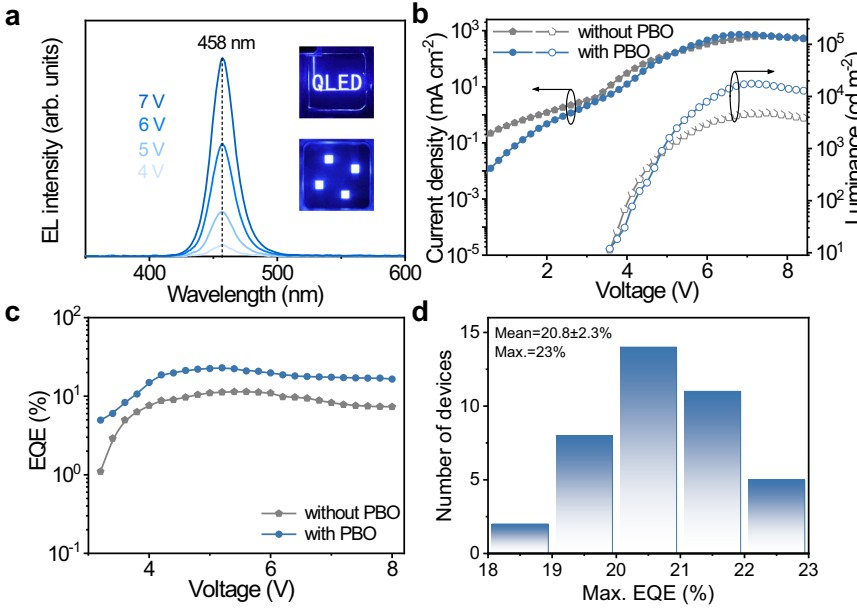

**Fig. 2 | QD-LEDs characterizations. a** Electroluminescence (EL) spectra of QD-LEDs with PBO under different driving voltages, and the inset shows the photograph of EL emission from the device operated at 6 V. **b** The current density-luminance-voltage characteristics and **c** external quantum efficiency (EQE) as a function of voltage for the QD-LEDs without and with PBO. **d** Statistical histogram of EQE for 40 devices with PBO inserting layer.

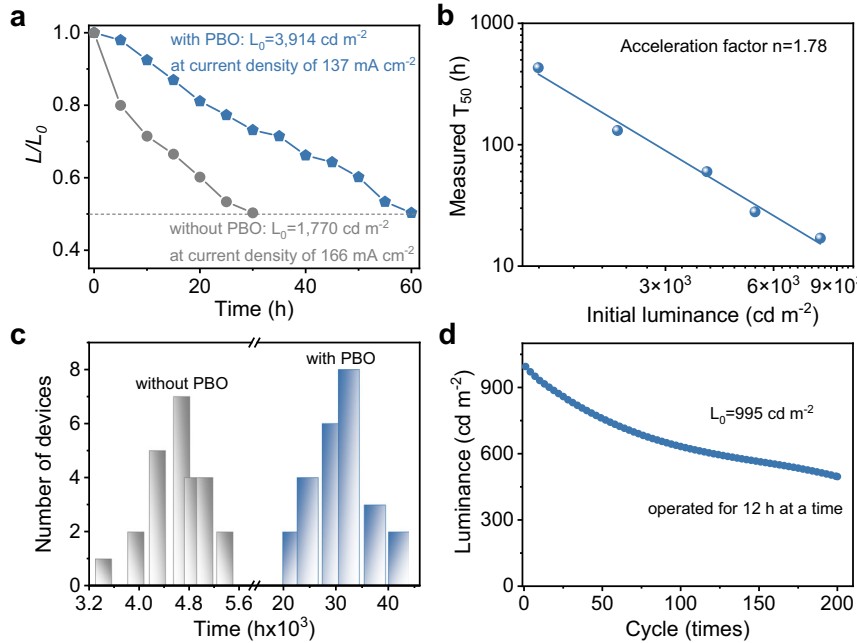

**Fig. 3 | Stability characterizations. a** The operational lifetimes (actual luminance/initial luminance ($L/L_O$) versus time) of QD-LEDs without and with PBO. **b** Extrapolation of accelerating factor (n) for the lifetime estimation by fitting the Log($T_{50}$)-Log($L_0$) data points. The initial luminance ($L_O$) and the measured time (T) for certain luminance degradation follow the empirical formula: $L_0^n \cdot T$ = constant.

Here, n is defined as the accelerating factor, which can be obtained by fitting the values at multiple initial luminances. In our devices, the accelerating factor is about 1.78. **c** Statistical histograms of the lifetime of 25 devices without and with PBO. **d** The operational cyclability of the device with PBO operated for 12 h at a time.

obtained in QD-LED without PBO. More importantly, the device has good operation cycle stability, and the brightness ($L_0$ = 995 cd m$^{-2}$) reduced to 50% after ~200 cycles (each cycle lasts 12 h; Fig. 3d).

### Mechanism of mitigating hole accumulation

To further investigate the origins of enhanced performance enabled by PBO, we measured the capacitance-voltage characteristics (Fig. 4a). The capacitances of both devices begin to increase as the applied voltage surpasses 2.8 V, indicating charge carriers begin to accumulate at the interfaces of PBO/QD and TFB/QD. As the applied voltage increases up to 3.5 V, the capacitances start to decrease, indicating the capacitors in TFB/QD interface have been broken down as a result of sufficient hole injection. The device with PBO has smaller capacitances in the whole voltage range, suggesting reduced charge accumulations. To test whether the charge accumulation happens at the HTL/QD interface, we further measured the electro-absorption (EA) spectra of both types of device (Fig. 4b, at 0.5 V). We can distinguish the EA signal of different layers, which indicates the electrical field distribution, according to their characteristic absorption peaks. The electrical field gets larger in the QD layer but smaller in the HTL when we introduce PBO at the HTL/QD interface, suggesting larger carrier concentrations in the QD layer but reduced hole accumulation in the HTL. The reduced hole accumulation in the HTL can prevent the oxidation of HTL, while higher charge carrier concentration in the QD layer increases the recombination probability.

To find out the main cause of device degradation, we measured the EA spectra and tr-EL under various biases for the fresh (F), degraded devices (after 30 min operation under 5.0 V bias) (D30 min) and degraded devices with power off after 24 h (P24 h) (Supplementary Figs. 6 and 12). From the EA signals from HTL (416 nm) and QDs (458 nm) (Fig. 4c), we observed a reduction in the electric field of HTL while the electric fields of QDs are unchanged for both two devices, indicating the device degradation is related to the damage of HTL rather than QDs. The reduction of the EA signal from HTL is smaller when PBO is introduced to the device (Fig. 4d), consistent with the

observed longer lifetimes. Furthermore, the unchanged EA spectra and tr-EL for degraded devices with power off after 24 h indicate the damage of HTL is irreversible.

In summary, by introducing PBO as a buffer layer between the QD layer and the HTL in QD-LEDs, we fabricated high-performance pure blue QD-LEDs with a maximum EQE of 23% and state-of-the-art long $T_{95}$ operational lifetime of more than 5200 h, corresponding to $T_{50}$ operational lifetime of more than 41,000 h hours at 100 cd m$^{-2}$. The capacitance-voltage characteristics and EA spectra indicated that the introduction of PBO not only enhances the efficiency of hole injection and alleviates electron leakage but also reduces the damage to the HTL. Both the high efficiency and device stability in this work exceed those of all previously reported solution-processed pure blue QD-LEDs (23% vs. 20.2%, and 41,000 h vs. 15,850 h). This work offers valuable insights into the design of high-efficiency and stable pure blue QD-LED.

## Methods

### Chemicals

Cadmium oxide (CdO, 99.998%, powder), zinc oxide (ZnO, 99.9%, powder), sulfur (S, 99.998%, powder), oleic acid (OA, 90%), 1-octanethiol (OT, 98%), 1-octadecene (ODE, 90%), zinc acetate (Zn(OA)$_2$, 99.99%), dimethyl sulfoxide (DMSO, 99.7%), tetramethylammoniumhydroxide (TMAH, 98%), chlorobenzene (99%), magnesium acetate tetrahydrate (Mg(OAc)$_2$·4H$_2$O, 99.98%), zinc(II) acetate dihydrate (Zn(OAc)$_2$·2H$_2$O, 99.99%), paraffin oil, n-Octane (99%), ethanol (99.8%) and N,N-Dimethylformamide (DMF, 99.89%) were purchased from Aldrich. Paraffin oil (analytical grade), hexanes (analytical grade), acetone (analytical grade), isopropanol (analytical grade), and methanol (analytical grade) were obtained from Beijing Chemical Reagent Co. Ltd, China. TFB was purchased from American Dye Source, Inc. All materials were used as received.

### Synthesis of CdZnS/ZnS core/shell quantum-dots

The pure blue emitting CdZnS/ZnS QDs have been synthesized by a typical synthetic procedure[25], 0.3 mmol of CdO, 0.1 mmol of ZnO,

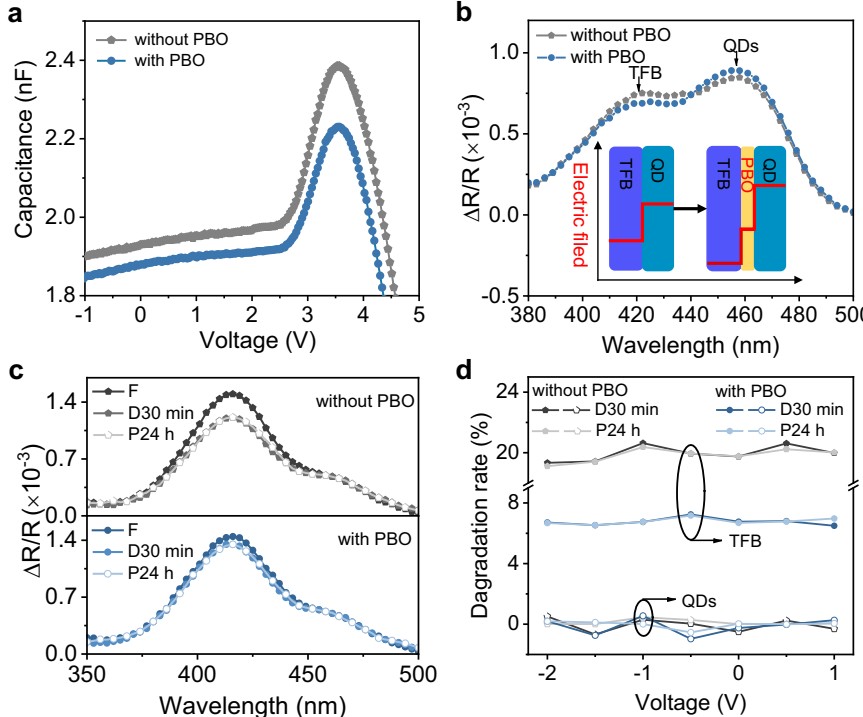

**Fig. 4 | Performance enhancement mechanism investigations. a** The capacitance-voltage characteristics of QD-LEDs without and with PBO. **b** Electro-absorption (EA) spectra for the devices under 0.5 V bias without and with PBO, which reflects the changes in the electric field distribution for the device without and with PBO as the inset shows. **c** Bias-dependent EA signals of TFB and QDs before degradation (F), after degradation (turn on for 30 min) (D30 min), and power off for 24 h after degradation (P24 h) for the device without and with PBO, respectively. **d** The difference in the Stark signal of TFB before, and after degradation and 24 h after degradation for the device without and with PBO, respectively.

15 mL of paraffin oil and 1 mL of OA were placed in a 100 mL round flask. The mixture was degassed at 150 °C for 20 min, and then the solution was heated to 300 °C under $N_2$ flow with vigorous stirring to form a clear mixture solution of $Cd(OA)_2$ and $Zn(OA)_2$. At this temperature, 0.4 mL sulfur stock solution was quickly injected into the reaction flask. After the core growth for 10 min, ZnS shell was over-coated by the following procedures: a desired amount of $Zn(OA)_2$ and octanethiol (1.2 equivalent amounts refer to $Zn(OA)_2$ diluted in 5 mL ODE) was dropwise into the reaction solution at a rate of 6 mL h$^{-1}$ using a syringe pump, and then further annealed at 310 °C for 30 min. After the reaction was completed, the temperature was cooled down to room temperature and the QDs were purified using acetone or methanol.

### Synthesis of ZnMgO nanoparticles

The colloidal ZnMgO nanoparticles (NPs) were synthesized by using a solution-precipitation process according to the method reported previously with some modification[2]. For a typical synthesis, 15 mL of $Mg(OAc)_2·4H_2O$ and $Zn(OAc)_2·2H_2O$ solution in DMSO (0.5 mol L$^{-1}$) and 5 mL of TMAH solution in ethanol (0.55 mol L$^{-1}$) were mixed and stirred for 1 h under ambient air, in which the mass fraction of $Mg(OAc)_2·4H_2O$ was 12.5%. Subsequently, the as-obtained products were washed and precipitated by adding ethanol and n-hexane twice. Finally, the resulting ZnMgO NPs were dispersed in ethanol at a concentration of 20 mg mL$^{-1}$.

### Fabrication of quantum-dot light-emitting diodes

The devices were fabricated using patterned indium tin oxide (ITO) glass substrates. These substrates were ultrasonically cleaned with detergent, deionized water, acetone, and isopropanol for 15 min, respectively, and then treated with UV ozone for 15 min. The mixed solution (1:1) of PEDOT:PSS (poly(3,4-ethylenedioxythiophene)/

poly(styrenesulfonate), AI 4083) and isopropanol used as hole-injection layers (HILs), was spin-coated onto the ITO substrates at the spin speed of 5500 rpm and baked at 150 °C for 15 min in air. Then, these substrates were immediately transferred to the $N_2$-filled glove box for spin-coating of TFB, PBO, QDs and ZnMgO, which were used as HTLs, buffer layer, EMLs and ETLs, respectively. TFB was spin-coated at a concentration of 6 mg mL$^{-1}$ in chlorobenzene (3000 rpm for 50 s), and then baked at 150 °C for 30 min. Next, the PBO solution in DMF was spin-coated at 3000 rpm for 30 s and baked at 120 °C for 30 min. This procedure was repeated to obtain different thicknesses of PBO layers. Subsequently, the pure blue CdZnS/ZnS QDs solution in n-octane was then spin-coated at 2000 rpm for 40 s, forming a 32 nm active layer. In turn, a 50 nm-ZnMgO layer was deposited on QDs by spin-coating its ethanol solution with a concentration of 20 mg mL$^{-1}$ at 2000 rpm for 30 s, followed by baking at 80 °C for 30 min. Finally, an Al anode was deposited via thermal evaporation at a rate of ≈ 0.1 nm s$^{-1}$ under a high vacuum of $4 × 10^{-6}$ Torr. The overlap between the Al cathode and the ITO anode defines an active area of 4 mm$^2$.

### Characterization and instrumentation

In acetonitrile solution with 0.1 M tetrabutylammonium-perchlorate amine as the electrolyte, the voltammetric curves of TFB and PBO films were multiple cycles measured by a BAS 100B, the scanning speed was 40 mV s$^{-1}$ and the test voltage range was −2.0 V-1.5 V. The UPS was performed on a Thermo Scientific ESCALAB 250 XI equipment with a He I discharge lamp (hυ = 21.22 eV). The surface morphologies of TFB, TFB/PBO, TFB/QD, and TFB/PBO/QD films were analyzed using AFM (Dimension Icon). UV-vis absorption and PL spectra were measured by an Ocean Optics spectrophotometer (model PC2000-ISA). Transmission electron microscopy (TEM) studies were performed using a JEOL JEM-2010 electron microscope operating at 200 kV. The energy

dispersive spectroscopy (EDS) mappings were carried out by four symmetrically designed EDS detectors equipped on FEI Talos F200X. The cross-section images of the QD-LEDs were carried out using FEI Talos F200X TEM. The phase and the crystallographic structure of the QDs were investigated by X-ray diffraction (XRD, D8-ADVANCE). The current density-voltage characteristics of QD-LEDs were analyzed using an Agilent 4155 C semiconductor parameter analyzer with a calibrated Newport silicon diode under ambient conditions. The luminance was calibrated using a Photo Research spectroradiometer (PR735). The EL spectra were obtained with an Ocean Optics spectrometer (USB2000, relative irradiance mode) and a Keithley 2400 source meter. Capacitance-voltage measurements were carried out using an Agilent 4282 A precision LCR meter with a modulating frequency of 30 KHz. Electro-absorption spectroscopy: A white light is generated by a xenon lamp (Zolix, Gloria-X150A) and splits into a monochromatic parallel beam by a monochromator (Zolix, Omni-$\lambda$300i). The monochromatic parallel beam probes the sample through the ITO side and is reflected by the back electrode. A sinusoidal voltage with a frequency of 1 kHz was overlayed to the DC bias to modulate the internal electric field for low-noise detection, producing a periodical bias in the form of $V(t) = V_{DC} + V_{AC} \sin(\omega t)$, here $\omega$ is the modulating frequency. Negative DC biases were used to measure the electro-absorption spectra to avoid carrier injection into the devices. The back-reflected signal was collected by the photomultiplier tube (Hamamatsu, H10721-20). A transimpedance amplifier (Femto, DHPCA-100) and a lock-in amplifier (SR830) were connected to the detector and locked to 1st harmonic frequency for low-noise measurement. The final signal was the ratio of the signals with the modulation of $V_{AC}$ and the DC signal. To lower the noise, the time constant of the lock-in was set at 1 s. Each data point is averaged from 500 measurements. Tr-EL: The blue QLEDs were driven by a (Keysight, 33512B) with 1 kHz, 10 μs voltage pulse. The tr-EL signal was collected by a high-speed photomultiplier tube (Hamamatsu, H10721-20) and transformed into a current signal. The current signal was sent to a transimpedance amplifier (Femto, DHPCA-100) and then collected by a digital oscilloscope (Rigol Technologies, DS1000Z).

## Data availability

All data are available in the main text or the Supplementary Information and can be obtained upon request from the corresponding author (chenfei.henu@henu.edu.cn; ffj@ustc.edu.cn; shenhuaibin@henu.edu.cn). Source Data has been deposited in figshare under accession code (https://doi.org/10.6084/m9.figshare.24432031)[40].

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

## Acknowledgements

We gratefully acknowledge the financial support from the National Key R&D Program of China (Grant No. 2023YFE0205000), the National Natural Science Foundation of China (Grants No. 62204078, No. U22A2072, No. 52272167), and Key R&D Program of Jiangxi Province (Grant No. 20192BBE50062).

## Author contributions

H.S. and F.F. conceived the concept and supervised the project. W.Z., B.L. and C.C. contributed equally. W.Z., C.C., Q.L., J.Y. and F.C. fabricated the devices and collected the performance data of the QD-LEDs. L.W., F.W. and Y.C. synthesized the materials. B.L. performed Tr-EL and EA experiments. F.C., B.L., H.S. and F.F. wrote the manuscript. H.S., F.F., W.Z., F.C. and Q.Z. conducted data analysis. H.S., F.F. and Z.D. provided financial support. H.S., F.F. and F.C. directed the project. All authors contributed to the scientific discussion and modified the manuscript.

## Competing interests

The authors declare no competing interests.
