## [Peer Review File · Nature Communications]

Stable and Efficient Pure Blue Quantum Dot LEDs Enabled by Inserting an Anti-oxidation LayerREVIEWER COMMENTS

Reviewer #1 (Remarks to the Author):

please see attachement.

Reviewer #2 (Remarks to the Author):

The authors used PBO as an interlayer between the HTL and the QD layer to improve hole injection and block the electron leakage current. The devices exhibited the highest EQE up to 23% and T50 of >41,000 h at 100 nit. However, PBO has been reported in a number of QLEDs with similar functionality (e.g. Chem. Commun. 2019, 55, 3501; J. Phys.: Conf. Ser., 2020, 1637, 012066) and its role is readily known. Next, the authors claimed that their blue QLEDs emit "pure blue" different from the sky-blue ones in Reference 3, and thus the value is the highest among "pure blue" devices, although the lifetime value described in this article is lower than that in Reference 3 (Nat. Commun. 2023, 14, 284). I am not sure if this can be said to be the recordable value when compared in limited conditions. Finally, the EQE is considered to be the highest, but given that the previous record is ~22% (Nat. Photon. 2022, 16, 505), it is hard to say that this is a significant advance. The research committee must consider that Cd-free QLEDs have already demonstrated a comparably high EQE of 20.2%, published in Nature 2020, 586, 385, while this work uses Cd-based QDs. Therefore, due to lack of novelty and low impact on the research field, I do not recommend the manuscript for publication in Nature Communications.

Reviewer #3 (Remarks to the Author):

The article titled "Stable Pure Blue Quantum Dot LEDs with 23% External Quantum Efficiency Enabled by Inserting an Anti-oxidation Layer," authored by Wenjing Zhang et al., presents a pure blue QD-LED device that exhibits an exceptional external quantum efficiency (EQE) of 23% and an impressive T50 operational lifetime exceeding 41000 hours at the initial brightness of 100 cd m⁻². The authors applied an anti-oxidation layer of PBO, which led to an enhancement in the stability of TFB HTL. Based on these findings, I recommend the publication of this article in Nature Communications.

The incorporation of PBO demonstrated a positive impact on mitigating the oxidation of HTL and achieving improved energy leveling of the layers in QD-LED devices. The authors presented a comprehensive range of device performances to substantiate the reliability of their data. This approach clarified the effect of PBO on the operational aspects of the device. The authors supported their hypothesis with robust analyses involving cyclic voltammetry, UPS, current densities of LED devices, HOD, and EOD.

However, I am intrigued by the authors' assertion that the insertion of the PBO layer promotes hole injection, as mentioned on page 6: "We observed that in devices with an inserted PBO layer, the EL indeed rises faster than those without PBO (Fig. 1d and Supplementary Fig. 4), suggesting that the PBO layer promotes hole injection." I have reservations about this claim, as there appears to be a reduction in current density in the LED device, as depicted in Figure 2(d), and a decrease in HOD, as shown in Supplementary Figure 8(a). I hope the authors will scrutinize this aspect more thoroughly.

While PBO has shown promising effects on device performance, I believe there is still room for further enhancement when it comes to employing antioxidant layers. Is PBO the optimal choice for an antioxidant layer, or are there specific properties of such layers that require

careful control? I encourage the authors to explore progressive ideas in this regard. Additionally, I posit that the introduction of an interlayer in a device, leading to the suppression or facilitation of electrons or holes, could potentially impact temporary charging of the device rather than causing irreversible disruption. Therefore, I suggest that the authors delve deeper into discussing the enduring physical changes in HTL or QDs in devices both with and without the PBO layer. Furthermore, exploring device performance after the dissipation of the charge following a certain duration of operation could provide valuable insights.

In this paper, the author proposed to introduce an antioxidant transition layer (PBO) between the QD layer and HTL to mitigate the damage caused by hole accumulation in HTL, at the same time, the electron leakage is reduced and therefore the authors demonstrate high EQE. The referee found the cyclic voltammetry measurement represented in this paper is interesting, it provides meaningful insights into the degradation mechanism of QLED. Although inserting organic materials between HTL and QDs layers have been used before, the authors demonstrate a different strategy here: inserting an more oxidation resistive transition layer to absorb excessive holes to mitigate the oxidation induced degradation. Also, the introduction of PBO significantly enhanced the T50 lifetime from 4995 h to 41022 h ($T_{95}=87\text{h}$), representing a breakthrough in pure-blue ($<470\text{ nm}$) region.

In general, this is an interesting and informative paper, and I strongly recommend the publication on Nature Communications. Some issues need to be addressed:

1. Previous literature reported that the introduction of intercalation between ETL and QD is conducive to inhibiting the quenching effect of ETL defect state on QD fluorescence. In this paper, can PBO play this role?
2. The CBM of PBO (between TFB and QD) seems to act as a ladder of electronic leakage. Why can the introduction of PBO inhibit electronic leakage? The author should expand discussion a little to make it clear.
3. In Figure 1, the author tested the electrochemical stability of PBO materials, but how about the thermal stability of PBO materials? Does the annealing temperature during subsequent device preparation affect the stability of PBO materials?
4. As for TEM and EDS characterization of CdZnS/ZnS quantum dots in Supplementary Figure 3, resolution is not high. Could the authors provide clearer pictures as support? Moreover, the EDS provided cannot fully characterize the distribution positions of Cd, Zn, and S elements within the quantum dots, so can the author provide EDS (line scan) spectra of multi-element comparison?
5. In Figure 2(b), the reviewer found special marks such as a, b, c, and d, but no corresponding explanation was found in the paper. The authors need to supply or correct them.
6. The impact factor n in Figure 3(b) seems to be faulty, and the author needs to double-check whether the data is correct.
7. In Figure 1(d), how is the 2.9 eV bandgap of the QD emission layer determined?
8. A recently published paper (J. Phys. Chem. Lett. 2023, 14, 1777–1783) is suggested to be cited for the analyses of the charge injection process.

To conclude, this is an interesting and informative paper. If all my concern can be properly addressed, I believe it is of interests to the potential readers of Nature Communications.

Reviewer #1 (Remarks to the Author):

In this paper, the author proposed to introduce an antioxidant transition layer (PBO) between the QD layer and HTL to mitigate the damage caused by hole accumulation in HTL, at the same time, the electron leakage is reduced and therefore the authors demonstrate high EQE. The referee found the cyclic voltammetry measurement represented in this paper is interesting, it provides meaningful insights into the degradation mechanism of QLED. Although inserting organic materials between HTL and QDs layers have been used before, the authors demonstrate a different strategy here: inserting an more oxidation resistive transition layer to absorb excessive holes to mitigate the oxidation induced degradation. Also, the introduction of PBO significantly enhanced the T_{50} lifetime from 4995 h to 41022 h ($T_{95}=87$ h), representing a breakthrough in pure-blue (<470 nm) region.

In general, this is an interesting and informative paper, and I strongly recommend the publication on Nature Communications. Some issues need to be addressed:

1. Previous literature reported that the introduction of intercalation between ETL and QD is conducive to inhibiting the quenching effect of ETL defect state on QD fluorescence. In this paper, can PBO play this role?

Response: The PL, PL QY, and PL decay curves for QD films without and with PBO antioxidant layer were tested, as shown in Figure R1. The PL intensity of the QD film on TFB/PBO layers obviously increases compared with the QD film on TFB layers, and the PL QYs of the QD thin layers on glass, TFB film, and TFB/PBO film are $\approx 89\%$, $\approx 58\%$, and $\approx 71\%$, respectively. Also, the corresponding time-resolved PL showed that the exciton decay time of TFB/QD film decreases by 25% compared with QD film, while it increases to 8.41 ns by inserting the PBO layer, closing to that of QD film (10.51 ns). The above results indicate that the quenching of QD emissions by TFB HTL can be significantly suppressed by introducing the PBO layer.

Figure R1 (Supplementary Figure 9) | (a) PL spectra, (b) PL QY data, and (c) PL decay curves of QD, TFB/QD, TFB/PBO/QD film on glass substrate.

2. The CBM of PBO (between TFB and QD) seems to act as a ladder of electronic leakage. Why can the introduction of PBO inhibit electronic leakage? The author should expand discussion a little to make it clear.

Response: The electron migration between interfaces is not only related to the energy offset between HTL and QDs, but also to the energetic disorder of HTLs. The energetic disorder of HTLs is determined by static disorder and dynamic disorder, and the former is due to structural disorder (for example, variations in molecular packing and conjugation lengths of the segments) or defects. In this work, the energy offset between PBO and QDs is 1.05 eV, which can effectively block the leakage of electrons. Also, the introduction of PBO increases the energy level thickness of electrons from QDs to TFB, effectively suppressing the leakage of electrons. On the other hand, it can be seen from the molecular structure of TFB and PBO that TFB has two long C_8H_{17} branch chains, which increases the structural disorder. However, PBO molecules have a more rigid structure and fewer defects (Figure R1), reducing the energetic disorder and thus suppressing electron leakage.

3. In Figure 1, the author tested the electrochemical stability of PBO materials, but how about the thermal stability of PBO materials? Does the annealing temperature during subsequent device preparation affect the stability of PBO materials?

Response: As can be seen from the molecular structure, the heterocyclic aromatic polymer, PBO, which has a rigid chemical structure and large molecular weight, can offer many outstanding properties such as high heat resistance. Therefore, we supplemented the thermogravimetric (TGA) characterization of PBO material, as shown in Figure R2. According to the TGA data, the mass weight of PBO decreased

only by 3% when heated to 120 °C (the annealing temperature of the PBO layer), and even when the temperature was as high as 158 °C, the mass weight of PBO only decreased by 5%, demonstrating excellent heat resistance and stability. Besides, the annealing temperature of the QDs/ZnMgO layer after PBO is only 60 °C, so the subsequent fabrication of QLEDs will not affect the stability of PBO.

Figure R2 (Supplementary Figure 2) | Thermogravimetric (TGA) data analysis of PBO materials.

4. As for TEM and EDS characterization of CdZnS/ZnS quantum dots in Supplementary Figure 3, resolution is not high. Could the authors provide clearer pictures as support? Moreover, the EDS provided cannot fully characterize the distribution positions of Cd, Zn, and S elements within the quantum dots, so can the author provide EDS (line scan) spectra of multi-element comparison?

Response: We retested the TEM and EDS of CdZnS/ZnS QDs. As shown in Figure R3, the core/shell QDs have an average diameter of about 10 nm with zincblende crystal structure. From the energy dispersive spectroscopy (EDS) elemental mapping, the Cd atoms are mainly located at the QD cores and the S atoms are distributed throughout the core/shell regions. The Zn atoms exist in the core but are mainly distributed in the shell, corresponding to the element line scan spectra.

Figure R3 (Supplementary Figure 5) | QDs characterizations. (a) TEM image and histogram of diameters of 100 CdZnS/ZnS QDs. (b) XRD pattern of as-prepared QDs. The black line indicates the peak positions of standard references of bulk zinc-blende ZnS. (c) A HAADF image. (d)-(f) EDS elemental mapping of Zn, Cd and S elements from several CdZnS/ZnS core/shell QDs shown in c. (g) Elements line scan spectra in c. (h) Absorption and PL spectra of CdZnS/ZnS core/shell QDs. The inset is the photograph of QD-octane dispersion under UV irradiation.

5. In Figure 2(b), the reviewer found special marks such as a, b, c, and d, but no corresponding explanation was found in the paper. The authors need to supply or correct them.

Response: We have corrected the manuscript and deleted a, b, c, d, as shown in Figure R4.

Figure R4 (Figure 2) | QD-LEDs characterizations. (a) EL spectra of QD-LEDs with PBO under different driving voltages, and the inset show the photograph of EL emission from the device operated at 6 V. (b) The J-L-V characteristics and (c) EQE as a function of voltage for the QD-LEDs without and with PBO. (d) Statistical histogram of EQE for 40 devices based on PBO.

6. The impact factor n in Figure 3(b) seems to be faulty, and the author needs to doublecheck whether the data is correct.

Response: We have double-checked the data, and ran some tests again, as shown in the following (Figure R5).

Figure R5 (Figure 3) | Stability characterizations. (a) Operational lifetimes (luminance versus time) of QD-LEDs without and with PBO. (b) Extrapolation of accelerating factor (n) for the lifetime estimation by fitting the $\text{Log}(T_{50})\text{-Log}(L_0)$ data points. The initial luminance (L) and the measured time (t) for certain luminance degradation follow the empirical formula: $L_0^n \cdot T = \text{constant}$. Here, n is defined as the accelerating factor, which can be obtained by fitting the values at multiple initial luminance. In our devices, the accelerating factor is about 1.78. (c) Statistical histograms of the lifetime of 25 devices without and with PBO. (d) The operational cyclability of the device with PBO operated for 12 h at a time.

7. In Figure 1(d), how is the 2.9 eV bandgap of the QD emission layer determined?

Response: Thanks for the reviewer's reminder. We checked the data and corrected it, as shown in Figure R6.

Figure R6 (Supplementary Figure 1) | UPS measurement of CdZnS/ZnS core/shell QDs. (a) The Tauc plot of QDs between $(\alpha h\nu)^2$ and photon energy. The inset is the absorption of QDs. (b) UPS spectra of the high-binding energy secondary electron cutoff regions and the valence-band edge regions of QDs.

8. A recently published paper (J. Phys. Chem. Lett. 2023, 14, 1777-1783) is suggested to be cited for the analyses of the charge injection process.

To conclude, this is an interesting and informative paper. If all my concern can be properly addressed, I believe it is of interests to the potential readers of Nature Communications.

Response: Thank you for your suggestion. We have added this reference (J. Phys. Chem. Lett. 2023, 14, 1777-1783) to the analysis of the charge injection process. In this reference (J. Phys. Chem. Lett. 2023, 14, 1777-1783), authors analyzed the operation mechanism of QLED in the delay, rising, balance, and decay stages, which can be correlated to the charge accumulations, charge injection and recombination, charge release and recombination, respectively. The accumulated holes are most likely located between HIL and HTL, and the accumulated electrons are most likely located between QD and ETL, as shown in the following. In this work, it is our opinion that due to the deep valence band energy level of blue QDs, there is a large injection barrier between the QD and the HTL, causing remarkable hole accumulation in HTL. Therefore, the citation in this reference is more conducive to our understanding of the charge injection and recombination process.

At the same time, we are admiring the useful comments and suggestions from the honorable reviewer, which after fulfilment will further improve the quality of our manuscript.

Reviewer #2 (Remarks to the Author):

The authors used PBO as an interlayer between the HTL and the QD layer to improve hole injection and block the electron leakage current. The devices exhibited the highest EQE up to 23% and T_{50} of $>41,000$ h at 100 nit. However, PBO has been reported in a number of QLEDs with similar functionality (e.g. Chem. Commun. 2019, 55, 3501; J. Phys.: Conf. Ser., 2020, 1637, 012066) and its role is readily known.

Response: We agree with you that PBO has been used in previous literature, but they have been used ubiquitously between ETL/QD as electron blocking layer in QD-LEDs (Chem. Commun. 2019, 55, 3501), or used to reduce the surface roughness of the electrode (J. Phys.: Conf. Ser., 2020, 1637, 012066).

In our work, they are inserted between HTL/QD layer for a completely different reason - to take in some holes from HTL to mitigate its damage. 1. From cyclic voltammetry characterizations, we unveil for the first time that the main cause of degradation of blue QD-LED is due to the oxidation of HTL by accumulated holes; 2. We propose a new strategy to address this problem - that is inserting an anti-oxidation PBO layer to take in the hole from HTL to protect HTL from degradation, this strategy allows us to achieve unprecedented operational lifetime and EQE.

Next, the authors claimed that their blue QLEDs emit “pure blue” different from the sky-blue ones in Reference 3, and thus the value is the highest among “pure blue” devices, although the lifetime value described in this article is lower than that in

Reference 3 (Nat. Commun. 2023, 14, 284). I am not sure if this can be said to be the recordable value when compared in limited conditions.

Finally, the EQE is considered to be the highest, but given that the previous record is ~22% (Nat. Photon. 2022, 16, 505), it is hard to say that this is a significant advance. The research committee must consider that Cd-free QLEDs have already demonstrated a comparably high EQE of 20.2%, published in Nature 2020, 586, 385, while this work uses Cd-based QDs. Therefore, due to lack of novelty and low impact on the research field, I do not recommend the manuscript for publication in Nature Communications.

Response: We agree with you that the device in the reference (Nat. Commun. 2023, 14, 284) and the reference (Nat. Photon. 2022, 16, 505) does have a long lifetime and high efficiency, but its EL peak is beyond 475 nm, which is sky-blue (> 466 nm). There is a large gap from the standard chromatic coordinates of pure blue (0.131, 0.046) in BT2020, as shown in Figure R7.

As far as we know, the commercialization of QD-LED is hindered by the short operational lifetime of blue color, any progress made in this direction, i.e. making blue QD-LED more stable, is critical to the future of QD-LED technology, because the QD-LED researches will get continuous support if this technology is commercialized, just like what is happening to OLED. We are afraid that, if the blue issue can't be properly addressed within the next several years, QD-LED can't be commercialized, and most of the QD-LED researchers, might have to change research direction. Herein, we reiterate that the main achievement of this work is the remarkably improved lifetime from 15850 h (Nature, 2020, 586, 385) to 41022 h (our work). This is an important progress on the most critical and challenging issue.

Regarding Cd-based or Cd-free based QDs, two leading companies, Samsung and TCL Co. have different strategies: the former one is focusing on Cd-free but the latter one is concentrating all efforts on Cd-based QDs. So far, it is hard to say which strategy is better, because the former one is more environmentally friendly, but the latter one is more feasible.

We admire the efforts and progress made reported by Kim et al. on Cd-free blue QDs, however, if the referee is aware of the fact that all QD synthesis, purification, and ligand exchange were performed in glove box, we believe the referee might also agree that Cd-free QD is still premature to be used for mass production. The reason behind it is simple, the ZnSeTe-based QDs are still prone to oxidation, and this is a tough problem that needs more efforts to address.

Figure R7 (Supplementary Figure 10) | CIE chromatic coordinates of references, BT2020 and our QD-LED.

Reviewer #3 (Remarks to the Author):

The article titled "Stable Pure Blue Quantum Dot LEDs with 23% External Quantum Efficiency Enabled by Inserting an Anti-oxidation Layer," authored by Wenjing Zhang et al., presents a pure blue QD-LED device that exhibits an exceptional external quantum efficiency (EQE) of 23% and an impressive T_{50} operational lifetime exceeding 41000 hours at the initial brightness of 100 cd m^{-2} . The authors applied an anti-oxidation layer of PBO, which led to an enhancement in the stability of TFB HTL. Based on these findings, I recommend the publication of this article in Nature Communications.

The incorporation of PBO demonstrated a positive impact on mitigating the oxidation of HTL and achieving improved energy leveling of the layers in QD-LED devices. The authors presented a comprehensive range of device performances to substantiate the reliability of their data. This approach clarified the effect of PBO on the operational aspects of the device. The authors supported their hypothesis with robust analyses involving cyclic voltammetry, UPS, current densities of LED devices, HOD, and EOD.

However, I am intrigued by the authors' assertion that the insertion of the PBO layer promotes hole injection, as mentioned on page 6: "We observed that in devices with an inserted PBO layer, the EL indeed rises faster than those without PBO (Fig. 1d and Supplementary Fig. 4), suggesting that the PBO layer promotes hole injection." I have reservations about this claim, as there appears to be a reduction in current density in the

LED device, as depicted in Figure 2(d), and a decrease in HOD, as shown in Supplementary Figure 8(a). I hope the authors will scrutinize this aspect more thoroughly.

Response: We have re-plotted the figure and revised the text to clarify that the hole injection enhanced with PBO insertion (Figure R8).

Since we observed increased EQE and reduced current density simultaneously with PBO insertion, the current efficiency loss through leakage has been depressed. In addition, by comparing the EOD and HOD with and without PBO insertion, we found the reduction in electron is more prominent (5 times) than the increase (2 times) in hole injection (Figure R8a), therefore, the overall current decreases under low driving voltage (Figure R8b).

Figure R8 (Figure 1 and Figure 2) | (a) J-V characteristic curves of HOD and EOD without and with PBO. (b) The J-V characteristics for the QD-LEDs without and with PBO.

While PBO has shown promising effects on device performance, I believe there is still room for further enhancement when it comes to employing antioxidant layers. Is PBO the optimal choice for an antioxidant layer, or are there specific properties of such layers that require careful control? I encourage the authors to explore progressive ideas in this regard.

Response: We agree with the referee that there is room for further enhancement of the antioxidant layer. As shown in Figure 4d, the introduction of PBO can reduce the damage of TFB during operation, but it cannot completely eliminate the damage of TFB. Therefore, the antioxidant layer can be further optimized. Under the current material

system, PBO is a relatively effective antioxidant material.

When selecting the antioxidant layer, firstly, we need to decide which functional layer of the QD-LED should be protected; moreover, in the case of introducing an additional layer of antioxidant material, the changes of the carrier accumulation and transport caused by the structure changes of QD-LED should be taken in to account.

Therefore, we need to screen the carrier mobility, energy level structure, and morphology of the antioxidant layer. Ensuring that the introduction of an antioxidant layer does not cause the weakening of carrier migration, and ideally, even increases the carrier concentration in the QD layer. Finally, the appropriate thickness of the antioxidant layer needs to be adjusted.

After confirming that TFB layer is the main cause of QD-LED attenuation, we select PBO the antioxidant layer (whose mobility is relatively high, valence band energy level is between TFB and QD, and the chemical stability is higher, Figure R9) that can effectively improve device stability and increase carrier concentration to improve the external quantum efficiency.

Figure R9 (Supplementary Figure 3, Figure 1, and Supplementary Figure 4) | (a)

The hole mobility of PBO, TFB, PVK, TCTA, CBP and TPD. (b) UPS spectra of the

high-binding energy secondary electron cutoff regions and the valence-band edge regions of TFB and PBO. The inset is the Tauc plot of TFB and PBO between $(\alpha h\nu)^2$ and photon energy. (c) The cyclic voltammetric curves of TFB and PBO. The insets are the chemical structures of TFB and PBO. (d) Schematic device structure, cross-sectional SEM image of QD-LED device.

Additionally, I posit that the introduction of an interlayer in a device, leading to the suppression or facilitation of electrons or holes, could potentially impact temporary charging of the device rather than causing irreversible disruption. Therefore, I suggest that the authors delve deeper into discussing the enduring physical changes in HTL or QDs in devices both with and without the PBO layer. Furthermore, exploring device performance after the dissipation of the charge following a certain duration of operation could provide valuable insights.

Response: We appreciate the referee's important and helpful comments. We believe that the electrochemical reaction during the operational process of QD-LED will cause irreversible damage to the device, and this damage mainly comes from the HTL (TFB). To further verify the problems mentioned by the referee, we repeated the electroluminescence over time of QD-LEDs, fluorescence lifetime, electroabsorption, and transient electroluminescence (tr-EL) for the degraded devices with power off after 24 h when the charge carriers were completely dissipated.

First, from the measurement of electroluminescence over time, we can conclude that the damage to devices is irreversible (Figure R10a). Then, the unchanged fluorescence lifetime of QD in device suggests that QD is not the main cause of device damage (Figure R10b).

The electroabsorption and tr-EL measurements for the fresh and degraded devices indicate the damage to TFB leads to the degradation of devices. Further, in repeated tests with power off after 24 h for the degraded devices, we found that the EA signal and rising edge of tr-EL had no change (Figure R10c and d). Therefore, we conclude that the damage occurs in TFB and it is irreversible.

Figure R10 | The performance mechanisms of devices without PBO and with PBO were investigated. (a) Electroluminescence over time of QD-LEDs. (b) PL decay curves of QD in devices. (c) Electroabsorption (EA) spectra under 0.5 V bias, and (d) tr-EL under 4.0 V bias for the device before degradation, after degradation (turn on for 30 min) and 24 h after degradation, respectively.

1. Bao, H. et al. Quantitative determination of charge accumulation and recombination in operational quantum dots light emitting diodes via time-resolved electroluminescence spectroscopy. *J. Phys. Chem. Lett.* **14**, 1777-1783 (2023).
2. Chen, X. et al. Blue light-emitting diodes based on colloidal quantum dots with reduced surface-bulk coupling. *Nat. Commun.* **14**, 284 (2023).
3. Deng, Y.Z. et al. Solution-processed green and blue quantum-dot light-emitting diodes with eliminated charge leakage. *Nat. Photonics* **16**, 505-511 (2022).
4. Kim, T. et al. Efficient and stable blue quantum dot light-emitting diode. *Nature* **586**, 385-389 (2020).

REVIEWERS' COMMENTS

Reviewer #1 (Remarks to the Author):

The authors have addressed the questions. I would like to support the publication as it is.

Reviewer #2 (Remarks to the Author):

Reviewers' questions were generally well answered.

I was wondering if PBO can increase lifetime for devices with green/red QDs.

Reviewer #3 (Remarks to the Author):

The authors have clearly elucidated all raised issues and, through supplementary experiments, have incorporated various concepts into the paper. Therefore, I support the publication of the work on Nature Communications.

Reviewer #1 (Remarks to the Author):

The authors have addressed the questions. I would like to support the publication as it is.

Response: We thank the referee very much for the inspirational comments!

Reviewer #2 (Remarks to the Author):

Reviewers' questions were generally well answered.

I was wondering if PBO can increase lifetime for devices with green/red QDs.

Response: Thank you again for your suggestion.

The efficiency and stability of blue QD-LEDs, particularly pure blue QD-LEDs, are far inferior to that of red and green devices, strongly hindering the commercialization of full-color QD-LED technology. These problems originate from the considerable hole injection barrier in QD-LEDs, which leads to severe hole accumulation and makes HTLs prone to oxidation. Therefore, in order to address this issue, an anti-oxidation layer (PBO (Poly-p-phenylene benzobisoxazole)) between the HTL and QD layer was inserted. The PBO transition layer in blue QD-LEDs takes in some holes from the HTL and itself is less prone to oxidation, allowing us to fabricate stable pure blue QD-LEDs. Meanwhile, thanks to its deeper HOMO level, hole injection is improved, leading to more efficient radiative recombination and improved peak EQE.

However, in red and green QD-LEDs, there are lesser injection barriers from holes due to their shallower HOMO. Therefore, there is much less hole accumulation between the HTL and QDs, and it is not necessary to introduce PBO. In addition, the introduction of PBO will increase the energy level thickness of holes from HTL to QDs, which, on the contrary, causes even worse device performance in red and green QD-LEDs.

Reviewer #3 (Remarks to the Author):

The authors have clearly elucidated all raised issues and, through supplementary experiments, have incorporated various concepts into the paper. Therefore, I support the publication of the work on Nature Communications.

Response: We thank the referee very much for the inspirational comments!